# Explaining the 'Trump Gap' in Social Distancing Using COVID Discourse

**Austin van Loon**
Stanford University
avanloon@stanford.edu

**Sheridan Stewart**
Stanford University
sastew@stanford.edu

**Brandon Waldon**
Stanford University
bwaldon@stanford.edu

**Shrinidhi K Lakshmikanth**
Stanford University
shrik@stanford.edu

**Ishan Shah**
Stanford University
ijshah@stanford.edu

**Sharath Chandra Guntuku**
University of Pennsylvania
sharathg@cis.upenn.edu

**Garrick Sherman**
University of Pennsylvania
garricks@sas.upenn.edu

**James Zou**
Stanford University
jamesz@stanford.edu

**Johannes Eichstaedt**
Stanford University
johannes.stanford@gmail.edu

## Abstract

Our ability to limit the future spread of COVID-19 will in part depend on our understanding of the psychological and sociological processes that lead people to follow or reject coronavirus health behaviors. We argue that the virus has taken on heterogeneous meanings in communities across the United States and that these disparate meanings shaped communities' response to the virus during the early, vital stages of the outbreak in the U.S. Using word embeddings, we demonstrate that counties where residents socially distanced less on average (as measured by residential mobility) more semantically associated the virus in their COVID discourse with concepts of fraud, the political left, and more benign illnesses like the flu. We also show that the different meanings the virus took on in different communities explains a substantial fraction of what we call the "Trump Gap," or the empirical tendency for more Trump-supporting counties to socially distance less. This work demonstrates that community-level processes of meaning-making determined behavioral responses to the COVID-19 pandemic and that these processes can be measured unobtrusively using Twitter.

## 1 Introduction

The COVID-19 Pandemic is one of the most significant and devastating events in human history, with over thirty-five million confirmed cases and over one million deaths globally (as of October 6, 2020). Part of the reason this virus has claimed so many lives is that individuals and the communities they are embedded in vary in the degree to which they take the virus seriously and follow suggested governmental guidelines aimed at reducing the spread of the virus, including the practice of social distancing. The degree to which the virus will continue to spread—especially as we await a second wave of infections—will depend not only on our biological understanding of the virus, but also on our understanding of the social factors that govern the degree to which everyday individuals are willing to and do in fact engage in coordinated efforts to help slow the spread of the virus.

An especially interesting case of response to the virus is the United States, the global leader in coronavirus deaths at the time of writing. Variation in response to the pandemic in the United States is particularly interesting because it does not appear to be randomly distributed, but in fact highly predictable. Political party identification in particular seems to be strongly associated with the degree to which individuals endorse health behaviors recommended by authoritative organizations such as the CDC and WHO (Allcott et al. 2020; Painter and Qiu 2020).

We seek to build on this work by identifying, at least in part, the reason *why* we observe these behavioral disparities. Rejecting these health behaviors is not *definitionally* related to identifying as a Republican or supporting Donald Trump. We contend instead that in addition to the virus's objective properties, SARS-CoV-2 has taken on heterogeneous *socially constructed meanings* (Berger and Luckmann 1966; Goldberg and Stein 2018), which vary over the population and shape communities' response to the pandemic. We further contend that this variation in how people understand the virus

partially accounts for the link between political party identification and these behavioral outcomes.

Research in sociology and anthropology (e.g. Geertz 1973; Rawlings and Childress 2019) finds that the meanings humans associate with different concepts tend to cluster in groups. One way that these meanings are clustered is by geography. Here, we leverage geographic variation in the content of discourse related to COVID-19 over the social media platform Twitter to explore the relationships between political identity, the potentially variable meaning of COVID-19, and social distancing.

To do this, we capture elements of what COVID-19 means to different populations using word embeddings, a technique that has been demonstrated to measure widely-held cognitive associations in groups (Bolukbasi et al. 2016; Caliskan et al. 2017; Garg et al. 2018; Kozlowski et al. 2019). This approach allows us to pick up on the disparate ways COVID-19 is interpreted by the residents of different U.S. counties. Further, we use Google Mobility Report data[1] to demonstrate that these disparate meanings both correlate with the degree to which individuals stay at home, our indicator of social distancing, and accounts for a substantial proportion of what we call the "Trump gap," or the tendency toward less social distancing within counties that exhibited greater support for Donald Trump in the 2016 presidential election.

First, using Word2Vec to capture interpretable, theoretically meaningful differences in meanings attributed to the virus, we find that counties in which the virus is discussed in a way that is semantically similar to the concepts (a) the political left, (b) fraudulence, and (c) the flu and the common cold are less likely to social distance. We further corroborate these findings with a minimal pair analysis, demonstrating that a BERT-based deep learning model trained on our data learns to associate social distancing with (a) blaming the pandemic on the political left, (b) questioning the reality of the virus, and (c) likening the virus to cold or flu. Finally, we demonstrate with mediation analysis that variation in the meanings associated with the virus mediates nearly one-fifth of the association between support for Donald Trump in the 2016 presidential election and social distancing at the county level.

## 2 Data

### 2.1 Twitter Corpus

We curated a corpus of English-language, U.S.-based tweets—any that contained at least one of a set of coronavirus-related hashtags—that were created between February 28 and May 18[2] (for other work using related corpora, see Eichstaedt et al. 2015; Jaidka et al. 2020). We used the subset of these tweets for which we could identify a U.S. county of origin and which were not retweets or shares. After pre-processing (see below), this resulted in a final corpus of approximately 1.1 million coronavirus-related tweets originating from 181 U.S. counties.

For our Word2Vec-based analyses (Sections 3 and 5), we preprocessed the Twitter data in several ways to ensure we were gaining the most accurate and potent signal possible. First, we removed stop words, URLs, punctuation, and all non-alphabetic characters and lowercased all letters. Next, we removed all tweets in which there were not at least two words. Finally, because we conduct analyses at the county level, we dropped all tweets originating in counties for which we had fewer than one-thousand tweets. In our BERT-based analyses (Section 4), we utilized a larger, less restrictive subset of the data that went through less preprocessing (described in Section 4).[3]

Using several publicly-available sources, we combined this data with county-level demographic information including income, population density, education, and Donald Trump's vote margin in each county during the 2016 U.S. presidential election.

### 2.2 Measuring Social Distancing

To capture adherence to social distancing guidelines at the county level, we use data from Google COVID-19 Community Mobility Reports, which record daily human mobility levels in various settings such as workplaces, residential areas, and retail/recreation. We focus on the change in mobility in residential areas, an indicator of the amount of people staying at home. In order for two individuals who do not live together to interact with one another, at least one of them must leave their home—

---

[1]https://www.google.com/covid19/mobility/

[2]We ended our observation period a week before the murder of George Floyd so that our outcome measure would not pick up any changes in mobility due to related protests.

[3]Due to an error in our Twitter scraping pipeline, tweets in our dataset are truncated to a maximum length of 140 characters. We plan to replicate our studies with non-truncated tweets in future work.

the exact behavior these data reflect.[4] For each county, we compute the average mobility score for the week leading up to the observation period (February 21 to February 27) as well as the average mobility score for the week just after the observation period (May 18 to May 24). We then take the latter less the former as our measure of social distancing adherence.[5]

The resulting measure tells us the degree to which residents of each county increased or decreased their adherence to social distancing guidelines over the observation period—specifically the degree to which residential mobility (which we interpret as being positively correlated to social distancing) increased over that time frame. In the counties we analyze in Sections 3 and 5, this measure ranges from 5.4 to 26.4 ($\mu = 14.5$, $\sigma^2 = 4.0$). Because this measure captures within-county variation over time, the potential for county attributes that do not vary over time (and whose relationship to the outcome do not vary over time) to confound our analyses is minimized.

## 3 Theory-Driven Word2Vec Analysis

We measure semantic associations between words as they are used in a county's COVID discourse using the Word2Vec word embedding algorithm (Mikolov et al., 2013). The algorithm places each word that appears in a corpus in a high-dimensional space where the proximity of two words in that space is proportional to the similarity of the words that appear in the linguistic contexts of those words. More intuitively, the Word2Vec algorithm creates a $k$-dimensional space in which semantically similar words, defined roughly as the interchangeability of those words, appear closer to one another than do semantically dissimilar words. We measure the semantic similarity of any two words by taking one less the cosine distance of the vector representations of the words' positions within the word embedding space.

Using Word2Vec to reliably measure *variation* in meaning is difficult because the algorithm requires a large amount of text to accurately represent the meaning of a word in any one context and because

spaces that result from applying the algorithm to disparate corpora do not necessarily align straightforwardly. Many existing solutions to these issues make difficult to verify assumptions about the process by which language is generated. We combat these issues with a novel, Word2Vec-based measurement strategy which maintains the algorithm's non-parametric properties.

This strategy, illustrated in Figure 1, first involves taking a fixed number of randomly selected tweets from each county and building a "baseline model." This baseline model leverages text from all counties to build a corpus large enough to train a reliable Word2Vec model and reflects semantic relations consistent across our corpus. Then, for each county, we fine-tune that baseline model to create a "county-specific model," which leverages a new sample of text from each county as well as the semantic information encoded in the baseline model to build an embedding space which reflects the county's potentially idiosyncratic semantic relationships. In order to make distances across different county-specific models more comparable, we measure the semantic similarity between words as the difference between the similarity of two keywords (e.g., "coronavirus" and "hoax") in the county-specific model less the similarity of the same word pair in the baseline model. This approximates measuring how counties deviate from the central tendency in their semantic association between words.

Imagine a scenario in which we have a population of one hundred counties, and we are interested in the degree to which residents of each county semantically associate the words "sky" and "blue." Further imagine that in ninety-nine of the one-hundred counties, these words are strongly associated and similarly so, but that in one county these words are much more weakly associated.[6] Our hope with this measurement strategy is that the semantic similarity between "sky" and "blue" in the baseline model (which again is built from a balanced random sample of texts from each county) would mostly reflect the ninety-nine counties in which "sky" and "blue" have relatively high semantic similarity, but that the county-specific models, each of which is fine-tuned on all text produced by its corresponding county, would more faithfully reflect each county's own, potentially unique, se-

---

[4] We acknowledge that this measure has severe limitations; it would not perfectly reflect, for instance, the degree to which individuals in a county traveled to residential areas that were not their own home.

[5] Each individual measure is in comparison to a pre-COVID baseline, but since we take the difference in these scores, we essentially "net out" this baseline.

[6] Perhaps the sky in that county has temporarily become orange/red due to wildfires in that area.

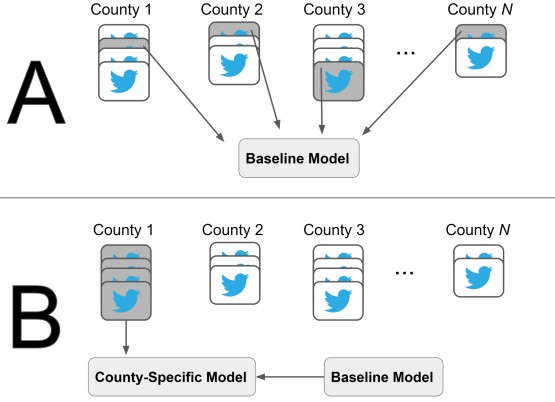

Figure 1: To build a baseline model and accompanying county-specific models, we (A) select a fixed-size, random sample of tweets from each county to train the baseline model and (B) create a copy of that baseline model for each county and fine-tune that baseline model on a sample of tweets from each model's respective county.

mantic relationships. Then, the difference between the semantic similarity of two words in the baseline model and any county-specific model would reflect how the county's own semantic relationships differs from the central tendency of all counties. In this scenario, we would expect a relatively small, positive deviation from the central tendency for the semantic similarity of "sky" and "blue" for ninety-nine of the one-hundred counties, and a much larger, negative deviation from the central tendency for the one outlier county. This strategy allows us to reliably capture variation in the degree to which different counties semantically associate these two words, even though we might not have a sufficient amount of text data to train a from-scratch Word2Vec model for any particular county.

To combat the stochasticity inherent to the Word2Vec algorithm as well as in sampling texts to build the baseline model, we create fifty separate baseline model/county-specific model pairs using random samples of one-thousand tweets from each county and take the average delta similarity score across all fifty. Finally, in accordance with related literature, we measure semantic relationships between concepts using a set of related words. The concepts of interest to us include the coronavirus ("coronavirus" and "covid"), the political left ("democrat", "democrats", and "liberals"), relatively benign illnesses ("flu", "influenza", and "cold"), and fraudulence ("hoax", "fake", and "scam"). We standardize (mean center and rescale

to have a standard deviation of one) each of these measures to ease interpretation.

### 3.1 Results

Table 1 presents the results of a series of ordinary least squares (OLS) regression models predicting a county's change in residential mobility from various linguistic associations of interest. In models one, two, and three, we test whether our measured linguistic associations are correlated with our measure of the change in social distancing at the county level. Then, in models four, five, and six, we test whether these same associations are robust to the inclusion of a set of competitive control variables, including Trump's vote margin, population density, median income, and education.

Model one reports a significant, negative relationship between our social distancing measure and the semantic similarity of the virus and the political left, which we believe reflects the degree to which individuals blame the virus on the political left. Model four shows that this association remains significant in the presence of our control variables. Model two demonstrates that counties social distances less the more they semantically associated the virus and the concept of fraudulence, or explicitly (and potentially implicitly) questioned the reality of the global pandemic. Model 5 shows that this association is robust to our control variables. Finally, model three shows that counties social distanced significantly less the more their COVID discourse exhibited a stronger association between COVID-19 and less serious illnesses, or the more they likened the coronavirus to these relatively benign illnesses. However, as can be seen in model six, this association is not robust to our controls.

In the models without controls, a one-standard-deviation increase in any of the three linguistic measures is associated with a sizeable decrease in social distancing (between -0.18 standard deviations and -0.22 standard deviations). Adding Trump vote margin and our control variables, we still see significant decreases in social distancing associated with two of our distance measures: a one-standard-deviation change in the association of the virus with the political left or with fraud is associated with a decrease in the change in social distancing of approximately 0.07 standard deviations and 0.09 standard deviations, respectively. Notably, a one-standard-deviation increase in Trump vote

Table 1: Meanings of COVID-19 and Changes in Social Distancing

|                          | (1)       | (2)       | (3)       | (4)         | (5)         | (6)         |
|--------------------------|-----------|-----------|-----------|-------------|-------------|-------------|
| Association with left    | -0.210**  |           |           | -0.0733*    |             |             |
|                          | (0.0737)  |           |           | (0.0303)    |             |             |
| Association with fraud   |           | -0.176*   |           |             | -0.0870**   |             |
|                          |           | (0.0742)  |           |             | (0.0299)    |             |
| Association with flu     |           |           | -0.221**  |             |             | -0.0420     |
|                          |           |           | (0.0735)  |             |             | (0.0303)    |
| Trump vote margin        |           |           |           | -0.416***   | -0.417***   | -0.435***   |
|                          |           |           |           | (0.0349)    | (0.0343)    | (0.0344)    |
| Other controls           |           |           |           | ✓           | ✓           | ✓           |
| Constant                 | 0.0       | 0.0       | 0.0       | 0.0         | 0.0         | 0.0         |
| $N$                      | 178       | 178       | 178       | 175         | 175         | 175         |
| $R^2$                    | 0.044     | 0.031     | 0.049     | 0.853       | 0.855       | 0.849       |

"Other controls" includes log population density, log median income, and education index

All measures are mean-centered and standardized

Standard errors in parentheses; excluded for constants

[+] $p < 0.10$, [*] $p < 0.05$, [**] $p < 0.01$, [***] $p < 0.001$

margin is associated with a decrease in our social distancing measure of approximately 0.4 standard deviations in models four, five, and six.

The $R^2$ statistics of models (1) - (3) are low by most standards in machine learning. Our goal is not to build a model that accounts for as much variation in the outcome as possible, but to use statistical modeling to rigorously test whether reliable associations between theoretically meaningful variables exist, which our models succeed in doing. In other words, we're not concerned whether these single, potentially noisy socio-psychological measures explain a sizeable proportion of the variance of our outcome variable, which is undeniably the product of myriad factors, but are instead concerned with whether these measures, which hopefully correspond to our theoretical constructs of interest, are related to our outcome in a consistent, reliable way.

## 4 Minimal Pair Analysis

Observational results like the ones reported in Table 1 have their strengths but also have inherent limitations. One major concern is that our Word2Vec model might pick up on information we do not intend for it to learn, introducing bias in the parameter estimates of our OLS regressions. One potential source of bias is that while the language produced by county residents might vary in the linguistic feature of interest (e.g., likening the virus to the flu), it might also vary on many other dimensions. The root of this problem is that we have neither complete information about nor experimental control over the text we collected from Twitter.

To partially address this, we train a deep-learning model to predict social distancing in a county from tweets originating from that county, then feed synthetic texts into that model that we systematically manipulate to only vary on the linguistic feature of interest. We assess whether the model has "learned" to associate social distancing with the same linguistic features we explored in Section 3, i.e. likening the virus to the flu, alleging that the pandemic is a hoax, or implicating the U.S. political left in the spread of the disease.

### 4.1 Predictive Model

We utilize Bidirectional Encoder Representations from Transformers (BERT) models (Devlin et al., 2018), which are highly general in that they have been shown to achieve state-of-the-art performance on a large array of Natural Language Processing tasks. We employ a BERT model that is pre-trained on masked language modeling and next-sequence prediction tasks and apply a fine-tuning approach with this model to a wide sample of our county-level Twitter corpus.

We set up the BERT fine-tuning as a sequence prediction task: given a sample of tweets from a county, the model predicts the same Google mobility metric described in Section 2.2 above. In contrast to analyses in Sections 3 and 5, in this section we drop only counties for which we have fewer than 512 tokens of Twitter data, leaving a total of 745 counties.

Moreover, for this section the original tweets are lowercased but are otherwise unaltered so as to leverage pre-trained BERT's understanding of syntactic dependencies and compositional semantic meaning (that is, we do not employ the pre-processing described above): we leave in non-ascii letters, urls, hashtags, and an anonymized user tag (<user>). A dummy token—[NEWTWEET]—is prepended to each tweet in the corpus. We concatenate the tweets of each county into documents which serve as the basis of the analysis.

## 4.2 Prediction Details and Evaluation

We use the `simpletransformers` library[7] to interface with the Huggingface transformers module (Wolf et al., 2019). We employ the `bert-base-uncased` model (12 transformer layers, 12 self-attention heads, hidden size 768, 110 million total parameters). The pooled output of the model is fed to a linear layer, with a mean squared error loss function to support regression. We employ a maximum sequence length of 512 (the longest sequence permitted by the model), a training batch size of 8, an evaluation batch size of 8, a learning rate of 4e-5 with Adam optimization (Adam epsilon value of 1e-8), and a training time of 10 epochs.[8]

A 10% sample of counties are left out of training for evaluation. We use the $R^2$ statistic to assess the accuracy of predicted values for the held-out set against their observed values.

We fine-tune and compute predictions on the held-out set with the `bert-base-uncased` model 30 times, generating 30 separate fine-tuned models. In each instance of fine-tuning, we first randomly sample a single 512-token sequence from each concatenation of tweets from every county in

the training set. In other words, each fine-tuned model is trained on distinct data consisting of 673 sequences of 512 tokens, one sequence for each county in the training set. A new single random 512-token sequence is also sampled for counties in the evaluation set when generating predictions for fine-tuned models on the evaluation set (that is, no fine-tuned model sees the exact same Twitter data when making predictions on the evaluation set). We average the predictions of each model on the held-out set to yield the predictions which we compare against our observed values.

## 4.3 Prediction Results

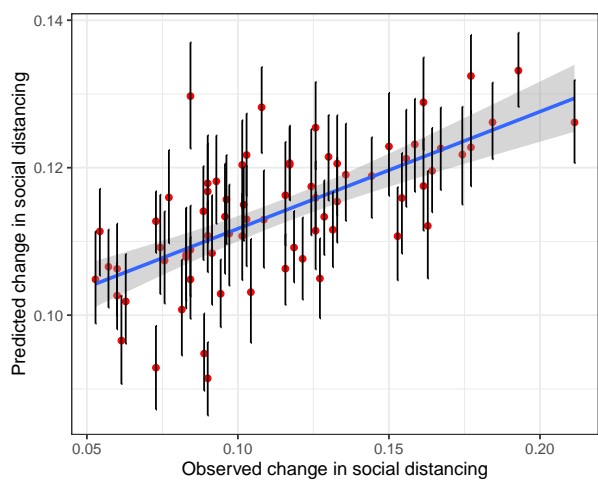

Figure 2: Average predicted change in social distancing (Google mobility metric, described in Section 2.1) from 30 fine-tuned BERT models, plotted against observed values in the evaluation set. Error bars indicate 95% confidence intervals of prediction means (displayed in red).

Figure 2 displays the predictions generated by the 30 fine-tuned BERT models against observed values in the evaluation set. The mean of these predictions achieves an $R^2$ of 0.43 on the held-out set, indicating that our BERT model is reasonably capable of predicting social distancing at the county level from tweets appearing in that county.

## 4.4 Experiments

Though we have demonstrated that our deep-learning model can reasonably predict our social distancing variable at the county level from Twitter data, the "black-box" nature of neural network-based language models precludes us from understanding what the model has "learned" about the

---

[7]https://github.com/ThilinaRajapakse/simpletransformers

[8]We determined the epoch size by analyzing the evaluation loss over epochs for one random fine-tuning, discarded for future analyses. We found that most reduction in loss occurred after 5 epochs and that the loss was mostly level for the 5 subsequent epochs. The learning rate, batch size, and Adam epsilon value are defaults for the `simpletransformers` package.

linguistic correlates of social distancing via examination of model parameter values alone.

One means of assessing what a neural network model has learned in training is minimal pair analysis. This analysis involves constructing artificial pairs of text, where members of the pair differ along some linguistic feature of interest. Each member of the pair is then fed separately to the model, which computes a prediction of the outcome for each member. Differences between predicted values for the two texts in each pair is evidence of a learned association between the linguistic feature and the outcome.

For our minimal pair analysis, we construct artificial pairs of synthetic Twitter data and retrieve our (fine-tuned and un-tuned) models' social distancing predictions for each. We compare the predicted differences within pairs produced by the tuned models *against* the predicted differences produced by the un-tuned model. This allows us to assess whether—over the course of fine-tuning—our neural network models learned the linguistic associations we measure in our theory-driven Word2Vec analyses (Section 3).[9]

Our minimal pairs differ according to one of the three following linguistic features of interest: blaming the pandemic on the political left, likening SARS-CoV-2 to the flu, and questioning the reality or severity of the pandemic. For each comparison, we first create a *control* document of Twitter data, consisting of five manually-constructed tweets that suppress the linguistic feature of interest; and an *experimental* document, where we manually alter tweets in the control document in order to evoke that linguistic feature. Table 2 presents a sample control and experimental tweet for each linguistic feature of interest—the full set of such texts that comprise the control and experimental documents are reported in the appendix. These documents are fed to our 30 fine-tuned BERT models as well as our un-tuned model to produce the predicted values of interest.

| *Likening the virus to the flu*: |
| **Control:** This virus is very different from the flu. |
| **Experimental:** This virus is very similar to the flu. |
| *Alleging fraudulence*: |
| **Control:** The pandemic is real. |
| **Experimental:** The pandemic is fake. |
| *Implicating the political left*: |
| **Control:** People need to start taking responsibility during this pandemic. |
| **Experimental:** Democrats need to start taking responsibility during this pandemic. |

Table 2: Example control and experimental sentences from the BERT experiments.

## 4.5 Experiment Results

Model predictions on the control versus experimental sets of data are displayed in Table 3. For each pair of data, we compare the average difference in predictions produced by our 30 fine-tuned models against the difference in predictions of the `bert-base-uncased` model initialized without fine-tuning (identical architecture and random seed as our fine-tuned models).

The fine-tuned models predict lower reductions in mobility for experimental documents vs. control documents on all three tests. By contrast, in the case of the "flu" and "fraudulence" experiments, the un-tuned model predicts a much *higher* reduction in mobility between our control and experimental data, counter to our theoretical predictions. Moreover, though the un-tuned model generates the predicted effect in the case of the "left" experiment, the magnitude of the effect is greater in the case of our fine-tuned models.

To assess whether the predictions of the fine-tuned models differed significantly from those of the un-tuned model in each of our three experiments, we used a one-sample Fisher randomization test to determine the probability that the observed difference in predicted social distancing for the control and experimental tweets produced by the untrained model is drawn from the same population as that same value for the 30 trained models[10] Each comparison is significant at the $p < 0.05$ level. These results demonstrate that our neural network models learned the linguistic correlates of social distancing that we explored in section 3 and corroborate the results presented therein.

---

[9]We take inspiration from the minimal pair analysis employed by Schuster et al. (2019), who probe the behavior of neural networks trained to predict the strength of pragmatic inference licensing in corpus data. For other recent applications of this type of analysis to neural network language modeling, see e.g. Ettinger et al. (2018), who probe the ability of neural nets to learn information about compositional semantic meaning; and Futrell et al. (2019), who probe their ability to learn syntactic representations.

---

[10]We use one-sided p-values since these analyses are meant

**Likening to the flu**\***

| Model | Control | Exper. | % Change |
|---|---|---|---|
| Fine-tuned | 0.130 | 0.127 | -2.22 |
| Un-tuned | 0.303 | 0.308 | 1.79 |

**Alleging fraudulence**\***

| Model | Control | Exper. | % Change |
|---|---|---|---|
| Fine-tuned | 0.169 | 0.167 | -1.06 |
| Un-tuned | 0.287 | 0.308 | 7.34 |

**Implicating the political left**\*

| Model | Control | Exper. | % Change |
|---|---|---|---|
| Fine-tuned | 0.190 | 0.165 | -12.9 |
| Un-tuned | 0.324 | 0.290 | -10.6 |

Table 3: (Average) predicted values for our three sets of control and experimental documents from our un-tuned model (fine-tuned models). Significance values assess the differences in differences using a one-sample Fisher randomization test. * $p < 0.05$; *** $p < 0.001$

## 5 Mediation analysis

We have shown that variation in the meanings attributed to the virus are associated with social distancing using two different methodological approaches. Now we test whether meanings attributed to the virus mediate the relationship between Trump vote margin in the 2016 election and social distancing.

As we note above, Trump vote margin at the county level in the 2016 presidential election is strongly associated with an increase in social distancing in the early days of the COVID-19 pandemic. Indeed, when regressing our measure of social distancing on Trump vote margin only, we get an in-sample $R^2$ of 0.38. The exact set of mechanisms linking Trump vote margin to social distancing, however, remains unknown. Mediation analysis allows us to test individual mechanisms, i.e. whether a given variable (the mediator) "explains" the relationship between the other variables. Put differently, mediation analysis seeks to identify whether part of the "total effect" of an independent variable on the outcome is accounted for by an "indirect effect," or an effect on the mediating variable which subsequently affects the outcome.

We use mediation analysis to formally test whether measures of the meanings attributed to the

to confirm the analyses presented in section 3.

virus mediate (explain) the relationship between Trump vote margin and social distancing in whole or in part. Rather than use our three hypothesized associations (Section 3), however, we acknowledge the complexity of what COVID-19 means and take an exploratory approach. First, we identify the one-thousand most frequent unigrams in the entire corpus, excluding "coronavirus" and "covid." Next, for each county, we measure the distance between each of these words and our linguistic indicators of the virus (the tokens "coronavirus" and "covid") as in Section 3. This gives us a larger set of associations between the virus and frequent words, providing more leverage on what meanings are attributed to the virus and how these vary from county to county. This in total creates a $1000 \times N$ matrix, where $N$ is the number of counties we analyze.

Next, we reduce the dimensionality of this matrix by estimating its top three principal components, creating three orthogonal measures that explain the most variation in these one-thousand measures. These three dimensions succinctly describe the variation in meanings attributed to the virus in each county in our analysis. When regressing our social distancing measure on these dimensions simultaneously, each coefficient is statistically significant at the $p < 0.001$ level and the model has an in-sample $R^2$ of 0.36.

In order to test whether these measures of the meanings attributed to the virus mediate the relationship between Trump vote margin in the 2016 election and social distancing in the early days of the pandemic, we use structural equation modeling (SEM) and proceed in two stages. In the first stage, social distancing is regressed on Trump vote margin alone, providing an estimate of the total effect of Trump vote margin on social distancing. In the second, social distancing is regressed on Trump vote margin and all three principal components.

### 5.1 Results

Figure 3 reports the resulting estimates. The dashed horizontal line between Trump vote margin and social distancing is the total effect, i.e. the strength of the association between Trump vote margin and social distancing in a bivariate regression. This estimate indicates that a one-standard-deviation increase in Trump vote margin is associated with a decrease in social distancing of 0.614 standard deviations. The solid horizontal line indicates the remaining direct effect when the three principal com-

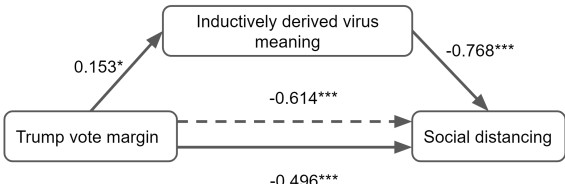

Figure 3: Results from structural equation modeling (SEM). Our inductively derived dimensions of virus meaning mediate 19.2% of the total effect of trump vote margin on our social distancing measure. All variables are mean-centered and standardized.

ponents are included. Pertinently, this direct effect is reduced to -0.496, meaning that a one-standard-deviation increase in Trump vote margin is associated with a decrease in social distancing of only 0.496 standard deviations net of our linguistically-derived measures.

The diagonal lines in Figure 3 represent the indirect effect of Trump vote margin on social distancing passed through the principal components (the mediating variables). We have summed the coefficients to simplify presentation, but it is difficult to interpret the coefficients corresponding to the principal components. We therefore focus on the portion of the total effect that is mediated by these measures of the meanings attributed to the virus. Consequently, we can conclude that our three components together mediate 19.2% of the association between Trump vote margin and social distancing.

## 6 Discussion and Conclusions

In our theory-driven analyses using Word2Vec, we demonstrated that counties social distanced less the more their COVID-19 discourse was indicative of cognitive associations between the virus and the concepts of (a) the political left, (b) fraud, and (c) less serious illnesses. We reaffirmed these findings using a deep learning model and BERT, showing that a model trained on our data learned to predict less social distancing for synthetic counties which (a) blame the political left for the pandemic, (b) question the reality or severity of the pandemic, and (c) liken the virus to the flu, compared to control versions of these counties. Through mediation analysis, we further showed that the heterogeneity in meanings the virus took on across the U.S. derived through PCA explains almost 20% of what we call the Trump gap, i.e. the empirical pattern that counties where residents supported Donald Trump in the 2016 election more social distanced

less.

While it is impossible to rigorously identify a causal effect with the present analyses, we mitigate concerns about spuriousness by using within-county variation in social distancing as well as by controlling for several potential confounding variables. Our experiments described in Section 4 further mitigate these concerns. Additionally, we establish temporal precedence by using the change in the outcome measured after the language variables. As always, omitted variables may be present; our analyses are suited to making strong claims about causality.

Overall, our results confirm that the different meanings attributed to the virus are associated with individuals' tendency to social distance. In fact, we demonstrate with mediation analysis that these meanings explain a great deal of the association between support for Trump in 2016 U.S. presidential election and social distancing, an empirical regularity that has been demonstrated (typically at the individual level) elsewhere. This means that if public health officials hope to increase adherence to social distancing (and potentially other health behavior-related) guidelines in the future, they must be mindful of the various meanings attributed to the virus in different communities, and attend to the dynamic process by which it might acquire new meanings.

This study outlines an approach to measuring the variation in meaning attributed to a novel concept in a population when limited text is available for each sub-population and builds on recent literature introducing minimal pair analysis of synthetic text. Additionally, our results demonstrate the importance of the process of the social construction of meaning, in accordance with arguments championed by sociologists for over a century. Finally, it demonstrates that large-scale social media text can be mined productively to recover traces of this collective meaning-making, and that, in principle, this can be done quickly enough to inform public health policy and messaging.

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

# 7 Appendix

## 7.1 Documents used in BERT minimal pair analysis

### 7.1.1 Mentioning the flu

**Control document:** [NEWTWEET] This virus is very different from the flu. [NEWTWEET] This virus is nothing like the flu. [NEWTWEET] This virus is much deadlier than the flu. [NEWTWEET] This virus is more dangerous than the flu. [NEWTWEET] I'm more afraid of this virus than I am of the flu.

**Experimental document:** [NEWTWEET] This virus is very similar to the flu. [NEWTWEET] This virus is just like the flu. [NEWTWEET] This virus is as deadly as the flu. [NEWTWEET] This virus is as dangerous as the flu. [NEWTWEET] I'm more afraid of the flu than I am of this virus.

### 7.1.2 Mentioning fraudulence

**Control document:** [NEWTWEET] The pandemic is real. [NEWTWEET] The pandemic needs to be taken seriously. [NEWTWEET] The virus is not a hoax. [NEWTWEET] The virus is not a

scam. [NEWTWEET] The pandemic is not at all made up.

**Experimental document:** [NEWTWEET] The pandemic is fake. [NEWTWEET] The pandemic doesn't need to be taken seriously. [NEWTWEET] The virus is a hoax. [NEWTWEET] The virus is a scam. [NEWTWEET] The pandemic is completely made up.

### 7.1.3 Mentioning the political left

**Control document:** [NEWTWEET] People need to start taking responsibility during this pandemic. [NEWTWEET] The behavior of some people during this pandemic is totally reckless! [NEWTWEET] I blame this virus on careless people. [NEWTWEET] People are the cause of this virus. [NEWTWEET] I'm so angry at the people who caused this pandemic!

**Experimental document:** [NEWTWEET] Democrats need to start taking responsibility during this pandemic. [NEWTWEET] The behavior of some democrats during this pandemic is totally reckless! [NEWTWEET] I blame this virus on careless democrats. [NEWTWEET] Democrats are the cause of this virus. [NEWTWEET] I'm so angry at the democrats who caused this pandemic!