# OpenReview forum: "Explaining the ‘Trump Gap’  in Social Distancing Using COVID Discourse"
_EMNLP/2020/Workshop/NLP-COVID — NLP-COVID19-EMNLP Oral_

### Official Review · AnonReviewer3 · 2020-09-22
**Interesting work on the correlation between communities’ response to the virus and their political association.**

**Rating:** 7
**Confidence:** 3

**Review:**

The paper presents an interesting analysis on the correlation between a population’s tendency to adhere to social distancing policy and the association of the virus with the political left, fraud and ultimately, individuals’ political party identification. Utilizing Google community mobility reports, the authors approximate the amount of people staying at home by change in their mobility in residential areas. The second part of the work supports the initial findings by demonstrating that (based on twitter data analysis) counties that tend to social distance less, are those that supported Donald Trump in the 2016 election.

I found this work interesting, well-structured and -written, and an enjoyable read overall. I have a few comments and clarification questions:

Section 2.1: the (presumed) positive correlation between residential mobility and social distancing needs to be better explained; it not entirely intuitive; yet, is one of the key assumptions of this work.

Building county-specific embeddings could be done using distributed representation for geographically situated language, as described in https://www.aclweb.org/anthology/P14-2134.pdf.

Table 1 needs more detailed interpretation IMO: say explicitly what are independent and dependent variables, what the numbers in the table represent (regression coeffs? pairwise correlations?). Also, the somewhat low R^2 in the first three models needs better interpretation.

Section 4.4: more details re the generation of an experimental document would be appreciated – *how* do you manually alter tweets in the control document? Some examples could help at this point.

Table 2: are the differences in control vs. experimental pair sets statistically significant?

---

As mentioned above, I enjoyed this work, and assuming authors’ clarifications re the above point in the final version, I’m recommending it for acceptance.

---

### Official Review · AnonReviewer1 · 2020-09-24
**Interesting work using tweets to analyse behavior patterns, would like to see more intuitive explanations**

**Rating:** 5
**Confidence:** 2

**Review:**

This paper studies the relationship between online language relating to COVID-19 and counrty-level behavior patterns in the United States.
The authors show that the variation in how people perceive the virus can reveal people's stance towards social distance measures, as well as towards Trump.

Strenths
* Novel way (Section 3) to use data from multiple sources, building associations between people online language and their stances or behavior patterns.

Limitations
* The writing of Section 4.4 and 4.5 is hard to understand, hard to connect these two sections and other sections. Suggest to provide more intuitive explanations.
* The section 5 is too brief. Details about data, method, setup are missing.


Questions and suggestions:
* Footnote 2, 5: period missing
* Suggest organize Section 2 into Section 2.1 explaning tweets and Section 2.2 explaning mobility data
* Page 3, 'we fine-tune that baseline model', do you mean 'continue training'?
* Move Appendix after references
* Suggest to add more intuitive explantions about the results, helping readers to understand these correlations.

---

### Official Review · AnonReviewer2 · 2020-09-25
**An interesting work that studies correlation between COVID discourses and community disparities**

**Rating:** 5
**Confidence:** 3

**Review:**

This work is interesting to read. This direction of study would offer insights to the community behaviors w.r.t. COVID-19, in addition to the related works that focus on Biomedical literature. However, justifications and explanations in this paper are very unfortunately insufficient.

1. The word "explain" appeared many times across the paper, but they should actually be "correlate" because the methodology used in this paper does not constitute an explanation.
2. Although the paper presents lots of technical details on neural models and text preprocessing, it lacks a proper explanation of how data are filtered, e.g. in the 3rd paragraph of Sec 3, how the use of the keywords liberals/democrats not presenting a bias in this study? Is keyword "republicans" also used? Further any control on the metioning of keywords and their community? (e.g. maybe democrats are mostly mentioned by republicans and vice versa?)
3. In the 2nd paragraph of Sec 3, how does having the county-specific model "combat" the large variation of word embeddings trained on small data? This is not explained at all.
4. In Table 1, what are the scales of those association numbers? BTW the table caption is weirdly formed and needs fix.